# Full State Constrained Flight Tracking Control for Helicopter Systems with Disturbances

**Yankai Li** [1,*,†] **, Yulong Huang** [1,*,†] **, Han Liu** [1] **and Dongping Li** [2]

1    School of Automation and Information Engineering, Xi'an University of Technology, Xi'an 710048, China
2    School of Sciences, Xi'an Technological University, Xi'an 710021, China
*    Correspondence: liyankai@xaut.edu.cn (Y.L.); hylxaut@foxmail.com (Y.H.)
†    These authors contributed equally to this work.

**Abstract:** In this paper, a full state-constrained anti-disturbance dynamic surface control method is proposed for six-degree-of-freedom unmanned helicopter systems under full state constraints and disturbances. Firstly, due to the underactuated characteristics of six-degree-of-freedom unmanned helicopter systems, an input–output feedback linearization method is used to transform the complex nonlinear systems into facilitated-control nonlinear ones. Based on the transformed systems, the nonlinear disturbance-observer-based control, backstepping control and Barrier Lyapunov function methods are used to construct the flight controller via uniting the state constraint control and dynamic surface control technologies. Then, Lyapunov stability theory is adopted for analysing the closed-loop tracking error systems, which confirms that the tracking errors are bounded under the proposed flight control scheme. Finally, a simulation in the MATLAB/Simulink environment verifies that the unmanned helicopter system can constrain all states under the action of the designed controller, with good dynamic performance.

**Keywords:** unmanned helicopter; nonlinear disturbance observer; backstepping control; state-constrained; dynamic surface control





## 1. Introduction

Unmanned aerial vehicles (UAVs) are common flying machines which are widely used in many fields of application, such as the military, agriculture, and others, and have demonstrated strong practical abilities. Indeed, UAVs can be divided into fixed-wing UAVs and unmanned helicopters according to their different flight principles. The unmanned helicopter has numerous advantages, including vertical take-off and landing, low-speed flying, fixed point hovering, and so on, compared to the fixed-wing UAV. Over the past few decades, with the development of composite materials, control theory, navigation and communication systems, unmanned helicopters have entered a new stage of development and received more and more attention [1,2]. However, unmanned helicopter systems have the characteristics of nonlinearity, strong coupling and being under-actuated, which make them vulnerable to the influence of disturbance during flight, which means the controller design becomes a difficult problem to overcome [3]. In recent years, many control methods have been used for the controller design of unmanned helicopter systems. In [4], an input-shaping and model-following control method was proposed to study the adaptive tracking control problem of helicopter hanged flight. In [5], the authors used the linear quadratic regulator control method to study the approach and landing of the helicopter on a limited platform, such as a moving vessel. In [6], the data-driven control problem was discussed for unmanned helicopters without model knowledge, in which the proposed control method was based on model-free adaptive control and the integration and variance approach, and the controller was designed using the input and output data. In [7], a control scheme was proposed by combining a state feedback linear controller and an $H_\infty$ controller for

reducing the adverse effects of helicopter icing conditions. The disturbance issues were mentioned for unmanned helicopter systems in parts of the above articles, which resulted in dynamic disturbances that can seriously affect the flight safety of helicopters. Therefore, it is necessary to study the advanced anti-disturbance control theory.

The anti-disturbance control is an important research direction in control theory domain. Based on the development of anti-disturbance control theory, many anti-disturbance control methods had been proposed, for example, the disturbance-observer-based control [8], active-disturbance rejection control [9] and $H_\infty$ robust control [10], and so on. Among these excellent anti-disturbance control technologies, the disturbance-observer-based control scheme is an effective method to reject the influence of disturbances, in which the disturbance observer is used to estimate the disturbances, whose estimation is applied in the feedforward controller, and ensures that the controller achieves good control performance. Hence, the disturbance-observer-based control scheme is widely used in various control systems. In [11], stochastic systems were investigated under multiple heterogeneous disturbances, which included parts of the known information disturbances, such as being $H_\infty$-norm bounded and induced by white noise, and a random disturbance observer was proposed via combining it with the $H_\infty$ control method to achieve disturbance suppression and attenuation. In [12], a shipboard platform system with unknown disturbance was studied, and the anti-disturbance controller was designed by uniting the nonlinear disturbance observer and vectorial backstepping technique. In [13], for the anti-disturbance attitude control problem of spacecraft, the extended state observer and backstepping control methods were employed to suppress disturbances. In [14], the author studied the anti-disturbance control problem of UAVs with unknown disturbance by using a sliding-mode control method. In [15], the disturbance-suppression problem of a space robot arm in a space environment was studied with the sliding-mode control method. Unmanned helicopters fly in the full-disturbances environment. Many disturbances affect the flight quality of helicopters, such as gusts, turbulence and wind shear. For these reasons, the anti-disturbance control is adopted based on the disturbance-observer-based control scheme to solve the robust control issue for helicopter systems. In our past works [16,17], the multiple random disturbances were discussed for helicopter nonlinear systems using composite anti-disturbance control schemes. Therefore, the anti-disturbance control issues are solved for helicopters, which would promote the control performance of the flight controller.

Furthermore, in order to further enhance the flying safety and stability for helicopters, it is necessary to constrain the intermediate control variables when accomplishing satisfactory performance and disturbance suppression ability. The constraint control technology has a wide range of applications in control theory, and there are many areas in practical engineering whose control variables need to be maintained within specified limits. In [18], a full state-constrained prescribed performance control problem was investigated through the adaptive fuzzy control method. In [19], the authors used the backstepping control and Barrier Lyapunov function methods to solve the state constraint problem of nonlinear switching systems. In [20], for the adaptive tracking control problem with asymmetric full state constraints, a control objective with bounded error was achieved by using the unified barrier function and the command filtered backstepping control methods. In [21], a state-constraint control problem of a class of electromagnetic active suspension systems was studied with a cotangent nonlinear state function with the neural network adaptive control method. In [22], a full state-constrained control problem for nonholonomic systems with parameter uncertainty was discussed via using the adaptive control method. In [23], the adaptive tracking control issue was studied for the interconnected nonlinear stochastic system with full state constraints. The above research results provide many new ideas for anti-disturbance control and state-constrained control, which are the new approach to improve the flying dynamic performance of helicopters.

Based on the above studies, in this paper, a flight-tracking controller is designed for six-degree-of-freedom unmanned helicopter systems with full state constraints and

disturbances. To enhance the practicability of the proposed controller, the intermediate control variables are constrained for the helicopter while the flight-tracking control task is achieved. Firstly, the six-degrees-of-freedom unmanned helicopter system is transformed by the input–output feedback linearization theory, and the tracking error system of the unmanned helicopter is obtained; secondly, a nonlinear disturbance observer is designed to estimate the system disturbance, and the disturbance estimate is used in the subsequent controller design; Then, in the process of controller design, the nonlinear disturbance observer, backstepping control, dynamic surface and Barrier Lyapunov function methods are combined to construct the full state-constrained flight-tracking control scheme. Under the conditions of full state constraints, the states of the tracking error systems are within the expected bounded range. Finally, the simulation is carried out in the MATLAB/Simulink environment to verify the effectiveness of the proposed flight controller.

This paper is organized as follows: in Section 2, we show the modeling process of helicopter systems; the full state-constrained anti-disturbance flight-tracking controller is constructed for helicopter systems in Section 3; the stability analysis is shown in Section 4 for helicopter tracking error systems; in Section 5, we give the numerical simulation to verify the validity of proposed flight control scheme; and Section 6 is the conclusions.

## 2. Unmanned Helicopter Models

According to the flight characteristics of an unmanned helicopter, the unmanned helicopter is considered as a six-degree-of-freedom rigid body. By considering the aerodynamic forces and moments during flying, the six-degree-of-freedom system models are established according to the Newton–Euler equations, which are shown as follows [17]:

$$
\begin{aligned}
\dot{\boldsymbol{P}}(t) &= \boldsymbol{V}(t) \\
\dot{\boldsymbol{V}}(t) &= g\boldsymbol{e}_3 + \frac{1}{m}\boldsymbol{R}(t)\boldsymbol{f}(t) + \boldsymbol{d}_1(t) \\
\dot{\boldsymbol{\Omega}}(t) &= \boldsymbol{H}(t)\boldsymbol{w}(t) \\
\boldsymbol{J}\dot{\boldsymbol{w}}(t) &= -\boldsymbol{w}(t)^{\times}\boldsymbol{J}\boldsymbol{w}(t) + \boldsymbol{\tau}(t) + \boldsymbol{d}_2(t)
\end{aligned}
\tag{1}
$$

where $\boldsymbol{P}(t) = [x(t)\ y(t)\ z(t)]^T$ denote the positions of the helicopter corresponding to the three coordinate axes, $\boldsymbol{V}(t) = [v_x(t)\ v_y(t)\ v_z(t)]^T$ denote the velocities of the helicopter, $\boldsymbol{\Omega}(t) = [\phi(t)\ \theta(t)\ \psi(t)]^T$ are the roll, pitch and yaw angle of the helicopter, respectively, and $\boldsymbol{w}(t) = [p(t)\ q(t)\ r(t)]^T$ are the three angular velocity components of the bodyframe coordinate system relative to the terrestrial coordinate system. $m$ and $g$ are the mass of helicopter and gravitational acceleration, $\boldsymbol{J} = diag\{j_{xx}\ j_{yy}\ j_{zz}\}$ is the inertia matrix of the helicopter, $\boldsymbol{d}_1(t)$ and $\boldsymbol{d}_2(t)$ denote the interference aerodynamic forces and moments of the helicopter during flight. $\boldsymbol{R}(t)$ denotes the rotation matrix of the bodyframe coordinate system relative to the terrestrial coordinate system:

$$
\boldsymbol{R}(t) =
\begin{bmatrix}
c_\theta c_\psi & s_\theta s_\phi c_\psi - c_\phi s_\psi & c_\phi s_\theta c_\psi + s_\phi s_\psi \\
c_\theta s_\psi & s_\phi s_\theta s_\psi + c_\phi c_\psi & c_\phi s_\theta s_\psi - s_\phi c_\psi \\
-s_\theta & s_\phi c_\theta & c_\phi c_\theta
\end{bmatrix}
$$

where $s_{(\cdot)}, c_{(\cdot)}$ denote $sin(\cdot)$ and $cos(\cdot)$, $\boldsymbol{H}(t)$ is attitude matrix:

$$
\boldsymbol{H}(t) =
\begin{bmatrix}
1 & \frac{s_\phi s_\theta}{c_\theta} & \frac{c_\phi s_\theta}{c_\theta} \\
0 & c_\phi & -s_\phi \\
0 & \frac{s_\phi}{c_\theta} & \frac{c_\phi}{c_\theta}
\end{bmatrix}
$$

$w(t)^\times$ is a cross-product matrix:

$$\boldsymbol{w}(t)^\times = \begin{bmatrix} 0 & -r(t) & q(t) \\ r(t) & 0 & -p(t) \\ -q(t) & p(t) & 0 \end{bmatrix}$$

$\boldsymbol{f}(t)$ and $\boldsymbol{\tau}(t)$ are the sums of external forces and moments on the centroid of the helicopter, respectively:

$$\boldsymbol{f}(t) = \begin{bmatrix} -s_a c_b T_M(t) \\ c_a s_b T_M(t) - T_T(t) \\ -c_a c_b T_M(t) \end{bmatrix} \tag{2}$$

$$\boldsymbol{\tau}(t) = \begin{bmatrix} z_m b(t) T_M(t) - z_t T_T(t) + c_{mb} b(t) \\ z_m a(t) T_M(t) + c_{ma} a(t) \\ x_t T_T(t) - Q_m(t) \end{bmatrix} \tag{3}$$

where $T_M(t)$ and $T_T(t)$ are main rotor thrust and tail rotor thrust, respectively, $a(t)$ and $b(t)$ are the longitudinal and lateral flapping angles, $c_{ma}$ and $c_{mb}$ are the main rotor pitch and roll moment intensity factors, $\boldsymbol{h}_m = [x_m\ y_m\ z_m]^T$ and $\boldsymbol{h}_t = [x_t\ y_t\ z_t]^T$ are the relative distances from the main rotor and tail rotor to the centroid of the helicopter on the bodyframe coordinate system, respectively. $Q_m(t) = c_{mq} T_M^{1.5}(t) + d_{mq}$ is the torque generated by the main rotor, where $c_{mq}$ and $d_{mq}$ are the main rotor torque factors.

The unmanned helicopter is flying with low velocity in the cruising flight phase. The flapping angles $a(t)$ and $b(t)$ are small, which satisfy $s_a \approx a(t)$, $s_b \approx b(t)$, $c_a = c_b \approx 1$, and in this case, we assume: $a(t)T_M(t) \approx 0$, $b(t)T_M(t) \approx T_T(t)$. Therefore, the sum of external forces $\boldsymbol{f}(t)$ is re-expressed as $\boldsymbol{f}(t) = [0\ 0\ -T_M(t)]^T$, and in this paper $[T_M(t)\ T_T(t)\ a(t)\ b(t)]^T$ is considered as the system input.

Before designing the controller, in order to simplify the controller design process, the following Assumption and Lemmas are necessary for building the flight controller, which are given as:

**Assumption 1** ([24]). *The disturbance aerodynamic forces and aerodynamic moments satisfy* $||\dot{\boldsymbol{d}}_1(t)|| < D_1, ||\dot{\boldsymbol{d}}_2(t)|| < D_2$, *where $D_1$ and $D_2$ denotes the unknown boundary.*

**Lemma 1** ([25]). *For any constant $\epsilon > 0$ and the appropriate dimensional vectors or matrices $\boldsymbol{X}$ and $\boldsymbol{Y}$, the following inequality holds:*

$$\boldsymbol{X}^T \boldsymbol{Y} + \boldsymbol{Y}^T \boldsymbol{X} \leqslant \varepsilon \boldsymbol{X}^T \boldsymbol{X} + \varepsilon^{-1} \boldsymbol{Y}^T \boldsymbol{Y}$$

**Lemma 2** ([26]). *For any constant $k_b$ and real variable $z(t)$, the following inequality holds while $z(t) < k_b$:*

$$\ln \frac{k_b^2}{k_b^2 - z^2(t)} \leqslant \frac{z^2(t)}{k_b^2 - z^2(t)}$$

## 3. Flight-Tracking Controller Design

### 3.1. System Transformation

In this paper, we design a state-constrained anti-disturbance flight controller for unmanned helicopter systems (1) and implement the asymptotic tracking of the expected tracking signals under the proposed controller. The control outputs are $\boldsymbol{P}_d(t)$ and $\psi_d(t)$, which denote the helicopter's expected position trajectories and yaw angle trajectory, respectively. In fact, the control inputs of systems (1) are $T_M(t)$, $T_T(t)$, $a(t)$ and $b(t)$, which are four-dimensional variables. Apparently, the helicopter systems (1) are the six-degree-of-freedom rigid body system models, which are typical underactuated nonlinear systems. In order to achieve the control objective, the input–output feedback linearization method is

used to obtain new equivalent systems via the systems (1). The flight-tracking controller is designed under the new systems.

Similar to the analysis process of literature [17], the systems (1) are extended to the following systems by adding two new variables:

$$
\begin{aligned}
\dot{\boldsymbol{P}}(t) &= \boldsymbol{V}(t) \\
\dot{\boldsymbol{V}}(t) &= g\boldsymbol{e}_3 - \frac{1}{m}\boldsymbol{R}(t)T_M(t) + \boldsymbol{d}_1(t) \\
\dot{T}_M(t) &= T_{M1}(t) \\
\dot{T}_{M1}(t) &= T_{M2}(t) \\
\dot{\boldsymbol{\Omega}}(t) &= \boldsymbol{H}(t)\boldsymbol{w}(t) \\
\boldsymbol{J}\dot{\boldsymbol{w}}(t) &= -\boldsymbol{w}(t)^\times \boldsymbol{J}\boldsymbol{w}(t) + \boldsymbol{\tau}(t) + \boldsymbol{d}_2(t)
\end{aligned}
\tag{4}
$$

The inputs and outputs of new systems (4) are $[T_{M2}(t) \ \boldsymbol{\tau}^T(t)]^T$ and $[\boldsymbol{P}_d^T(t) \ \psi_d(t)]^T$. Then, based on (4), a group of new variables are constructed from the systems (4) via using the input–output feedback linearization methods [17], which are given by:

$$
\begin{aligned}
\boldsymbol{x}_1(t) &= \boldsymbol{P}(t) - \boldsymbol{P}_d(t) \\
\boldsymbol{x}_2(t) &= \boldsymbol{V}(t) - \dot{\boldsymbol{P}}_d(t) \\
\boldsymbol{x}_3(t) &= g\boldsymbol{e}_3 - \frac{\boldsymbol{e}_3}{m}\boldsymbol{R}(t)T_M(t) - \ddot{\boldsymbol{P}}_d(t) \\
\boldsymbol{x}_4(t) &= -\frac{\boldsymbol{e}_3}{m}\boldsymbol{R}(t)T_{M1}(t) - \frac{\boldsymbol{e}_3}{m}\boldsymbol{R}(t)\boldsymbol{w}^\times(t)T_M(t) - \dddot{\boldsymbol{P}}_d(t) \\
\boldsymbol{x}_5(t) &= \psi(t) - \psi_d(t) \\
\boldsymbol{x}_6(t) &= \boldsymbol{\alpha}_1(t)\boldsymbol{w}(t) - \dot{\psi}_d(t)
\end{aligned}
\tag{5}
$$

where $\boldsymbol{\alpha}_1(t) = [0 \ \sin\phi(t)\sec\theta(t) \ \cos\phi(t)\sec\theta(t)]$. Moreover, new control inputs are constructed by using the input–output feedback linearization method according to $[T_{M2}(t) \ \boldsymbol{\tau}^T(t)]^T$ [17], which are given by

$$
\begin{aligned}
\boldsymbol{u}_1(t) &= \frac{T_M(t)}{m}\boldsymbol{R}(t)\boldsymbol{\beta}_1\boldsymbol{J}^{-1}\boldsymbol{w}(t)^\times \boldsymbol{J}\boldsymbol{w}(t) - \frac{2T_{M1}(t)}{m}\boldsymbol{R}(t)\boldsymbol{w}(t)^\times\boldsymbol{e}_3 - \frac{T_M(t)}{m}\boldsymbol{R}(t)\boldsymbol{w}(t)^\times\boldsymbol{w}(t)^\times\boldsymbol{e}_3 \\
&\quad - \frac{T_{M2}(t)}{m}\boldsymbol{R}(t)\boldsymbol{e}_3 - \frac{T_M(t)}{m}\boldsymbol{R}(t)\boldsymbol{\beta}_1\boldsymbol{J}^{-1}\boldsymbol{\tau}(t) - \boldsymbol{P}_d^{(4)}(t)
\end{aligned}
$$

$$
\begin{aligned}
u_2(t) &= -\boldsymbol{\alpha}_1(t)\boldsymbol{J}^{-1}\boldsymbol{w}(t)^\times \boldsymbol{J}\boldsymbol{w}(t) + \left[\frac{c_\phi}{c_\theta}\dot{\phi}(t) + \frac{s_\phi s_\theta}{c_\theta^2}\dot{\theta}(t)\right]q(t) - \left[\frac{s_\phi}{c_\theta}\dot{\phi}(t) - \frac{c_\phi s_\theta}{c_\theta^2}\dot{\theta}(t)\right]r(t) \\
&\quad + \boldsymbol{\alpha}_1(t)\boldsymbol{J}^{-1}\boldsymbol{\tau}(t) - \ddot{\psi}_d(t)
\end{aligned}
$$

Indeed, according to [17], $[\boldsymbol{x}_1(t) \ \boldsymbol{x}_2(t) \ \boldsymbol{x}_3(t) \ \boldsymbol{x}_4(t) \ x_5(t) \ x_6(t)]^T$ have practical physical significance, which are the helicopter position tracking errors, velocity tracking errors, accelerations generated by controllable join force, force variation rates generated by controllable join force, yaw angle tracking error and yaw angle rate error, respectively.

From (4) and (5), the new transformed systems are obtained, which are given as:

$$
\begin{aligned}
\dot{\boldsymbol{x}}_1(t) &= \boldsymbol{x}_2(t) \\
\dot{\boldsymbol{x}}_2(t) &= \boldsymbol{x}_3(t) + \boldsymbol{d}_1(t) \\
\dot{\boldsymbol{x}}_3(t) &= \boldsymbol{x}_4(t) \\
\dot{\boldsymbol{x}}_4(t) &= \boldsymbol{u}_1(t) - \frac{T_M(t)}{m}\boldsymbol{R}(t)\boldsymbol{\beta}_1\boldsymbol{J}^{-1}\boldsymbol{d}_2(t) \\
\dot{x}_5(t) &= x_6(t) \\
\dot{x}_6(t) &= u_2(t) + \boldsymbol{\alpha}_1(t)\boldsymbol{J}^{-1}\boldsymbol{d}_2(t)
\end{aligned}
\tag{6}
$$

where

$$\beta_1 = \begin{bmatrix} 0 & 1 & 0 \\ -1 & 0 & 0 \\ 0 & 0 & 0 \end{bmatrix}$$

According to the above process, in this paper, the flight-tracking controller is designed based on the systems (6), $u_1(t)$ and $u_2(t)$ are the control inputs of systems (6) and $x_1(t)$ and $x_5(t)$ are the outputs of systems (6). In what follows, the tracking controllers $u_1(t)$ and $u_2(t)$ are constructed such that the control outputs $x_1(t)$ and $x_5(t)$ converge to arbitrary bounded scopes, and the other states are constrained within reasonable domains.

### 3.2. Nonlinear Disturbance Observer Design

In order to suppress the influence of disturbance aerodynamic forces $d_1(t)$ and aerodynamic moments $d_2(t)$ on the helicopter systems, the nonlinear disturbance observers are used to estimate the disturbances, and substitute the disturbance estimates into the feedforward controller. For the disturbance aerodynamic forces $d_1(t)$, the disturbance estimator errors $\tilde{d}_1(t) = d_1(t) - \hat{d}_1(t)$ are defined and the disturbance observer is designed as follows:

$$\begin{aligned} \hat{d}_1(t) &= \delta_1(t) + L_1 x_2(t) \\ \dot{\delta}_1(t) &= -L_1\left[x_3(t) + \hat{d}_1(t)\right] \end{aligned} \tag{7}$$

where $\hat{d}_1(t)$ are the disturbance estimates, $L_1 = diag\{l_1, l_1, l_1\}$ is the disturbance observer gain matrix and $\delta_1(t)$ denotes the constructive function. The dynamics of the errors in the disturbance estimates are:

$$\begin{aligned} \dot{\tilde{d}}_1(t) &= \dot{d}_1(t) - \dot{\hat{d}}_1(t) \\ &= \dot{d}_1(t) - [\dot{\delta}_1(t) + L_1 \dot{x}_2(t)] \\ &= \dot{d}_1(t) - [-L_1 x_3(t) - L_1 \hat{d}_1(t) + L_1 x_3(t) + L_1 d_1(t)] \\ &= -L_1 \tilde{d}_1(t) + \dot{d}_1(t) \end{aligned} \tag{8}$$

Note that (4) and (6) are equivalent systems. The disturbance observer of $d_2(t)$ is designed on the basis of systems (4) as follows:

$$\begin{aligned} \hat{d}_2(t) &= \delta_2(t) + L_2 J W(t) \\ \dot{\delta}_2(t) &= -L_2\left[-w(t)^\times J w(t) + \tau(t) + \hat{d}_2(t)\right] \end{aligned} \tag{9}$$

where $\hat{d}_2(t)$ are the disturbance estimates, $L_2 = diag\{l_2, l_2, l_2\}$ is the disturbance observer gain matrix, and $\delta_2(t)$ denotes the constructive function. The dynamics of the errors in the disturbance estimates are:

$$\begin{aligned} \dot{\tilde{d}}_2(t) &= \dot{d}_2(t) - \dot{\hat{d}}_2(t) \\ &= \dot{d}_2(t) - [\dot{\delta}_2(t) + L_2 J \dot{w}(t)] \\ &= \dot{d}_2(t) - [-L_2 w(t)^\times J w(t) - L_2 \tau(t) - L_2 \hat{d}_2(t) - L_2 w(t)^\times J w(t) + L_2 \tau(t) + L_2 d_2(t)] \\ &= -L_2 \tilde{d}_2(t) + \dot{d}_2(t) \end{aligned} \tag{10}$$

Based on the designed disturbance observers, in what follows the disturbance estimates are introduced into the process of the state-constrained tracking controller.

### 3.3. State-Constrained Backstepping Controller Design

In this section, a full state-constrained anti-disturbance controller is designed based on backstepping control, and such that the expected trajectories of position $P_d(t)$ and yaw

angle $\psi_d(t)$ are tracked under the error state's constrained condition. The control flow graph of the unmanned helicopter is shown in Figure 1:

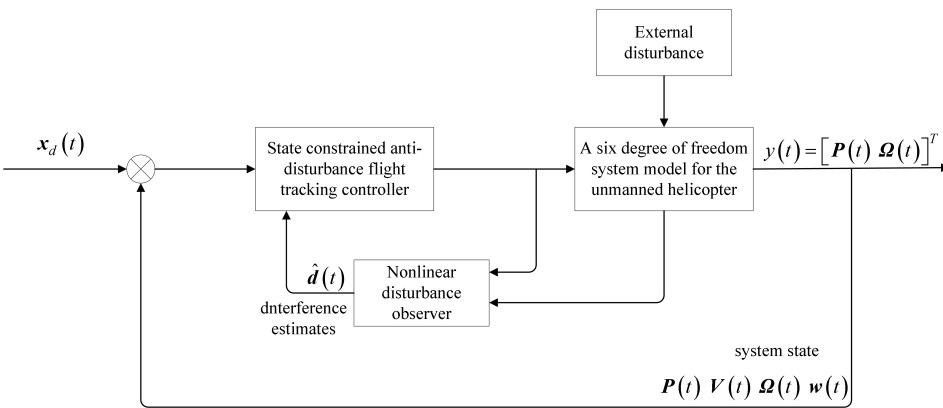

**Figure 1.** The position-tracking trajectories of the unmanned helicopter.

The control design mainly includes the design of a nonlinear disturbance observer and a composite anti disturbance tracking controller based on the combination of disturbance compensation and backstepping. The observer obtains disturbance estimates for feedforward compensation, handles the impact of disturbance on the closed-loop control system and derives a reference feedback controller based on the backstepping method to achieve the goal of asymptotically converging the system state. The two work together to achieve the effect of anti-disturbance tracking control. Even if the disturbance degrades to zero, the composite controller can maintain the performance of the reference feedback control and still maintain the asymptotic stability of the system.

Combining the backstepping control scheme with the dynamic surface control method, the first-order filters are introduced to estimate the virtual control inputs, and the problem of "differential explosion" is solved for the nonlinear controller. The specific design process is as follows:

Step 1: Define the tracking error as $e_1(t) = x_1(t)$. To ensure accurate tracking, the expected position $P_d(t)$ with adequate tracking control performance, the maximum helicopter position tracking error values are set as $k_{b1i}$, and $|e_{1i}(0)| \leqslant k_{b1i}$, for $i \in \{1, 2, 3\}$.

Define the Barrier Lyapunov function candidate $V_1(t)$ as:

$$V_1(t) = \frac{1}{2} \sum_{i=1}^{3} \ln \frac{k_{b1i}^2}{k_{b1i}^2 - e_{1i}^2(t)} \tag{11}$$

From (6), the derivative of (11) is given as

$$\dot{V}_1(t) = e_1^T(t) C_1(t) \dot{e}_1(t) = e_1^T(t) C_1(t) x_2(t) \tag{12}$$

where $C_1(t) = diag \left\{ \frac{1}{k_{b11}^2 - e_{11}^2(t)} \; \frac{1}{k_{b12}^2 - e_{12}^2(t)} \; \frac{1}{k_{b13}^2 - e_{13}^2(t)} \right\}$. To ensure the nonnegative definiteness of (12), the virtual control inputs $\alpha_1(t)$ are chosen as:

$$\alpha_1(t) = -k_1 A_1(t) e_1(t) - \frac{\varepsilon_1}{2} C_1(t) e_1(t) \tag{13}$$

where $k_1 \geqslant \frac{1}{k_{b1i}^2 - e_{1i}^2}$ for $i = 1, 2, 3$ is controller gain, $\epsilon_1 > 0$ is the positive constant, and $A_1(t) = C_1^{-1}(t)$.

To solve the "differential explosion" problem that arises during the backstepping controller designed process, a first-order filter with time constant $\lambda_1$ is constructed, and its outputs are substituted for the virtual control inputs $\alpha_1(t)$. The initial values of this

filter are the same as the initial values of the virtual control inputs. The filter is designed as follows:

$$\lambda_1 \dot{x}_{2c}(t) + x_{2c}(t) = \alpha_1(t), x_{2c}(0) = \alpha_1(0) \tag{14}$$

Define the errors between the filter outputs and the virtual control inputs as $\gamma_2(t) = x_{2c}(t) - \alpha_1(t)$. Then, the dynamic of $\gamma_2(t)$ is given by:

$$\dot{\gamma}_2(t) = \dot{x}_{2c}(t) - \dot{\alpha}_1(t) = -\frac{\gamma_2(t)}{\lambda_1} - \dot{\alpha}_1(t) \tag{15}$$

Step 2: Define the errors of the first dynamic surface as $e_2(t) = x_2(t) - x_{2c}(t)$. We choose the Barrier Lyapunov function candidate $V_2(t)$ as:

$$V_2(t) = \frac{1}{2} \sum_{i=1}^{3} \ln \frac{k_{b2i}^2}{k_{b2i}^2 - e_{2i}^2(t)} \tag{16}$$

where $k_{b2i} > 0$, for $i \in \{1,2,3\}$, are the maximum tracking errors of $e_{2i}(t)$. The derivative of (16) along with (6) is shown as:

$$\dot{V}_2(t) = e_2^T(t)C_2(t)\dot{e}_2(t) = e_2^T(t)C_2(t)\left[x_3(t) + d_1(t) + \frac{\gamma_2(t)}{\lambda_1}\right] \tag{17}$$

where $C_2(t) = diag\left\{ \frac{1}{kb_{21}^2 - e_{21}^2(t)} \ \frac{1}{kb_{22}^2 - e_{22}^2(t)} \ \frac{1}{kb_{23}^2 - e_{23}^2(t)} \right\}$.

Similar with the Step 1, in order to guarantee the nonnegative definiteness of $\dot{V}_2(t)$, the virtual control inputs $\alpha_2(t)$ are chosen as:

$$\alpha_2(t) = -k_2 A_2(t)e_2(t) - A_2(t)C_1(t)e_1(t) - \frac{\epsilon_2}{2}C_2(t)e_2(t) - \frac{\epsilon_3}{2}C_2(t)e_2(t) - \hat{d}_1(t) - \frac{\gamma_2(t)}{\lambda_1} \tag{18}$$

where $k_2 \geqslant \frac{1}{k_{b2i}^2 - e_{2i}^2}$ for $i = 1,2,3$ is the controller gain, and $\epsilon_2 > 0$ and $\epsilon_3 > 0$ are positive constants, $A_2(t) = C_2^{-1}(t)$.

We design the second first-order filter as:

$$\lambda_2 \dot{x}_{3c}(t) + x_{3c}(t) = \alpha_2(t), x_{3c}(0) = \alpha_2(0) \tag{19}$$

and define the errors between the filter outputs and the virtual control inputs as $\gamma_3(t) = x_{3c}(t) - \alpha_2(t)$

The dynamic of $\gamma_3(t)$ is given by:

$$\dot{\gamma}_3(t) = \dot{x}_{3c}(t) - \dot{\alpha}_2(t) = -\frac{\gamma_3(t)}{\lambda_2} - \dot{\alpha}_2(t) \tag{20}$$

Step 3: The error of the second dynamic surface is defined as $e_3(t) = x_3(t) - x_{3c}(t)$. The Barrier Lyapunov function candidate $V_3(t)$ is chosen as:

$$V_3(t) = \frac{1}{2} \sum_{i=1}^{3} \ln \frac{k_{b3i}^2}{k_{b3i}^2 - e_{3i}^2(t)} \tag{21}$$

where $k_{b3i} > 0$, $i = \{1,2,3\}$, are the maximum acceleration errors of the helicopter-controllable join force. The derivative of $V_3(t)$ along with (6) is given as:

$$\dot{V}_3(t) = e_3^T(t)C_3(t)\dot{e}_3(t) = e_3^T(t)C_3(t)\left[x_4(t) + \frac{\gamma_3(t)}{\lambda_2}\right] \tag{22}$$

where $C_3(t) = diag\left\{ \frac{1}{kb_{31}^2 - e_{31}^2(t)} \ \frac{1}{kb_{32}^2 - e_{32}^2(t)} \ \frac{1}{kb_{33}^2 - e_{33}^2(t)} \right\}$.

To ensure the nonnegative definiteness of $\dot{V}_3(t)$, the virtual control inputs $\boldsymbol{\alpha}_3(t)$ are chosen as:

$$\boldsymbol{\alpha}_3(t) = -k_3 \boldsymbol{A}_3(t) \boldsymbol{e}_3(t) - \boldsymbol{A}_3(t) \boldsymbol{C}_2(t) \boldsymbol{e}_2(t) - \frac{\epsilon_4}{2} \boldsymbol{C}_3(t) \boldsymbol{e}_3(t) - \frac{\boldsymbol{\gamma}_3(t)}{\lambda_2} \tag{23}$$

where $k_3 \geqslant \frac{1}{k_{b3i}^2 - e_{3i}^2}$ for $i = 1, 2, 3$ is controller gain, $\epsilon_4 > 0$ is a positive constant and $\boldsymbol{A}_3(t) = \boldsymbol{C}_3^{-1}(t)$.

We design the third first-order filter as:

$$\lambda_3 \dot{\boldsymbol{x}}_{4c}(t) + \boldsymbol{x}_{4c}(t) = \boldsymbol{\alpha}_3(t), \boldsymbol{x}_{4c}(0) = \boldsymbol{\alpha}_3(0) \tag{24}$$

and define the errors between the filter outputs and the virtual control inputs as $\boldsymbol{\gamma}_4(t) = \boldsymbol{x}_{4c}(t) - \boldsymbol{\alpha}_3(t)$. Then, the dynamic of this error $\boldsymbol{\gamma}_4(t)$ is shown as

$$\dot{\boldsymbol{\gamma}}_4(t) = \dot{\boldsymbol{x}}_{4c}(t) - \dot{\boldsymbol{\alpha}}_3(t) = -\frac{\boldsymbol{\gamma}_4(t)}{\lambda_3} - \dot{\boldsymbol{\alpha}}_3(t) \tag{25}$$

Step 4: The errors of the third dynamic surface are defined as $\boldsymbol{e}_4(t) = \boldsymbol{x}_4(t) - \boldsymbol{x}_{4c}(t)$. The Barrier Lyapunov function candidate $V_4(t)$ is chosen as:

$$V_4(t) = \frac{1}{2} \sum_{i=1}^{3} \ln \frac{k_{b4i}^2}{k_{b4i}^2 - e_{4i}^2(t)} \tag{26}$$

where $k_{b4i}$, $i \in \{1, 2, 3\}$ are the maximum force variation rate errors from the helicopter-controllable join force. The derivative of $V_4(t)$ along with (6) is given by:

$$\dot{V}_4(t) = \boldsymbol{e}_4^T(t) \boldsymbol{C}_4(t) \dot{\boldsymbol{e}}_4(t) = \boldsymbol{e}_4^T(t) \boldsymbol{C}_4(t) \left[ \boldsymbol{u}_1(t) - \frac{T_M(t)}{m} \boldsymbol{R}(t) \boldsymbol{J}^{-1} \boldsymbol{d}_2(t) + \frac{\boldsymbol{\gamma}_4(t)}{\lambda_3} \right] \tag{27}$$

where $\boldsymbol{C}_4(t) = diag \left\{ \frac{1}{kb_{41}^2 - e_{41}^2(t)} \ \frac{1}{kb_{42}^2 - e_{42}^2(t)} \ \frac{1}{kb_{43}^2 - e_{43}^2(t)} \right\}$.

To ensure the nonnegative definiteness of $\dot{V}_4(t)$, the virtual control inputs $\boldsymbol{u}_1(t)$ are chosen as:

$$\begin{aligned} \boldsymbol{u}_1(t) = &- k_4 \boldsymbol{A}_4(t) \boldsymbol{e}_4(t) - \boldsymbol{A}_4(t) \boldsymbol{C}_3(t) \boldsymbol{e}_3(t) - \frac{\epsilon_5}{2} \frac{T_M(t)}{m} \boldsymbol{R}(t) \boldsymbol{J}^{-1} \frac{T_M(t)}{m} \boldsymbol{J}^{-1} \boldsymbol{R}^T(t) \boldsymbol{C}_4(t) \boldsymbol{e}_4(t) \\ &+ \frac{T_M(t)}{m} \boldsymbol{R}(t) \boldsymbol{J}^{-1} \hat{\boldsymbol{d}}_2(t) - \frac{\boldsymbol{\gamma}_4(t)}{\lambda_3} \end{aligned} \tag{28}$$

where $k_4 \geqslant \frac{1}{k_{b4i}^2 - e_{4i}^2}$, for $i = 1, 2, 3$, is controller gain and $\epsilon_5 > 0$ is a positive constant, $\boldsymbol{A}_4(t) = \boldsymbol{C}_4^{-1}(t)$.

Step 5: Define the error term $e_5(t) = x_5(t)$ and define the Barrier Lyapunov function candidate $V_5(t)$ as:

$$V_5(t) = \frac{1}{2} \ln \frac{k_{b5}^2}{k_{b5}^2 - e_5^2(t)} \tag{29}$$

where $k_{b5} > 0$ is the maximum helicopter yaw angle tracking error. From (6), the derivative of is shown as:

$$\dot{V}_5(t) = \frac{e_5(t) \dot{e}_5(t)}{k_{b5}^2 - e_5^2(t)} = \frac{e_5(t) x_6(t)}{k_{b5}^2 - e_5^2(t)} \tag{30}$$

To ensure the nonnegative definiteness $\dot{V}_5(t)$, the virtual control input $\alpha_5(t)$ is chosen as:

$$\alpha_5(t) = -k_5 \left[ k_{b5}^2 - e_5^2(t) \right] e_5(t) - \varepsilon_6 \frac{e_5(t)}{k_{b5}^2 - e_5^2(t)} \tag{31}$$

where $k_5 \geqslant \frac{1}{k_{b5}^2 - e_5^2}$ is controller gain, and $\epsilon_6 > 0$ is a positive constant. The fourth first-order filter is designed as:

$$\lambda_4 \dot{x}_{6c}(t) + x_{6c}(t) = \alpha_5(t), x_{6c}(0) = \alpha_5(0) \tag{32}$$

Then, the error between the filter output and the virtual control input is defined as $\gamma_6(t) = x_{6c}(t) - \alpha_5(t)$ and the dynamic of $\gamma_6(t)$ is given by:

$$\dot{\gamma}_6(t) = \dot{x}_{6c}(t) - \dot{\alpha}_5(t) = -\frac{\gamma_6(t)}{\lambda_4} - \dot{\alpha}_5(t) \tag{33}$$

Step 6: The error of the fourth dynamic surface is defined as $e_6(t) = x_6(t) - x_{6c}(t)$ and the Barrier Lyapunov function candidate $V_6(t)$ is chosen as:

$$V_6(t) = \frac{1}{2} \ln \frac{k_{b6}^2}{k_{b6}^2 - e_6^2(t)} \tag{34}$$

where $k_{b6} > 0$ is the maximum helicopter yaw angle rate error. The derivative of $V_6(t)$ is:

$$\dot{V}_6(t) = \frac{e_6(t)\dot{e}_6(t)}{k_{b6}^2 - e_6^2(t)} = \frac{e_6(t)\left[u_2(t) + \boldsymbol{\alpha}_1(t)\boldsymbol{J}^{-1}\boldsymbol{d}_2(t) + \frac{\gamma_6(t)}{\lambda_4}\right]}{k_{b6}^2 - e_6^2(t)} \tag{35}$$

To ensure the nonnegative definiteness of $\dot{V}_6(t)$, the control inputs $u_2(t)$ are chosen as:

$$u_2(t) = -k_6\left[k_{b6}^2 - e_6^2(t)\right]e_6(t) - \frac{k_{b6}^2 - e_6^2(t)}{k_{b5}^2 - e_5^2(t)}e_5(t) - \frac{\varepsilon_7}{2\left[k_{b6}^2 - e_6^2(t)\right]}\boldsymbol{\alpha}_1(t)\boldsymbol{J}^{-1}\boldsymbol{J}^{-1}\boldsymbol{\alpha}_1^T(t)e_6(t)$$

$$- \boldsymbol{\alpha}_1(t)\boldsymbol{J}^{-1}\hat{\boldsymbol{d}}_2(t) - \frac{\gamma_6(t)}{\lambda_4} \tag{36}$$

where $k_6 \geqslant \frac{1}{k_{b6}^2 - e_6^2}$ is controller gain, and $\epsilon_7 > 0$ is the positive constant.

**Remark 1.** *As assumed in many other studies [27], $\Omega \subset D$ is a positive invariant set of the virtual control laws $\dot{\boldsymbol{\alpha}}_1(t), \dot{\boldsymbol{\alpha}}_2(t), \dot{\boldsymbol{\alpha}}_3(t), \dot{\boldsymbol{\alpha}}_5(t), V : D \to R$ is a continuous differentiable function satisfying $\dot{V}(\dot{\boldsymbol{\alpha}}_i) \leq 0$ for $i = 1, 2, 3, 5$ within $\Omega$, E is the set of all points within $\Omega$ satisfying $\dot{V}(\dot{\boldsymbol{\alpha}}_i) = 0$, and M is the maximal invariant set within E.*

**Remark 2.** *In this paper, an anti-disturbance flight controller is constructed for the helicopter systems via the disturbance observer, backstepping controller and BLF methods. During the helicopter flying, all of the tracking errors of helicopter systems are constrained in limitative scopes, which have many advantages for improving the dynamic performance of helicopter systems. Firstly, the helicopter would be safer under our proposed control scheme. For example, the variable $x_3(t)$ denotes the driving force acceleration of the helicopter, which is constrained in a limitative scope would avoid helicopter overturning. Secondly, the dynamic performance of helicopter would be better using our designed controller. For instance, the variable $x_1(t)$ is the position tracking error, which is constrained in a limitative scope would avoid the big overshoot, and enhance the flying performance.*

## 4. Stability Analysis

**Theorem 1.** *Consider the tracking error systems of unmanned helicopter (6) with disturbances, design the nonlinear disturbance observers (7) and (9), the virtual control laws (13), (18), (23) and (31), and tracking controller (28) and (36). For given some parameters $k_{bji} > 0$, $j = 1, 2, 3$ and $i = 1, 2, 3, 4$, $k_{bh} > 0$, $h = 5, 6$, the tracking errors of unmanned helicopter systems (6) converge to the arbitrary bounded scopes with full state constraints. If there exist some parameters $\epsilon_j > 0$, $j = 1, 2..., 13$, $k_i > 0$, $i = 1, 2..., 6$, such that the following conditions hold: $k_1 \geqslant \frac{1}{k_{b1i}^2 - e_{1i}^2}, k_2 \geqslant \frac{1}{k_{b2i}^2 - e_{2i}^2}, k_3 \geqslant \frac{1}{k_{b3i}^2 - e_{3i}^2}, k_4 \geqslant \frac{1}{k_{b4i}^2 - e_{4i}^2}, k_5 \geqslant \frac{1}{k_{b5}^2 - e_5^2}, k_6 \geqslant \frac{1}{k_{b6}^2 - e_6^2}, \bar{k}_7 = \frac{2}{\lambda_1} - \varepsilon_1^{-1} - \varepsilon_{10}, \bar{k}_8 = \frac{2}{\lambda_2} -$*

$\varepsilon_2^{-1} - \varepsilon_{11}, \bar{k}_9 = \frac{2}{\lambda_3} - \varepsilon_4^{-1} - \varepsilon_{12}, \bar{k}_{10} = \frac{2}{\lambda_4} - 2\varepsilon_6^{-1} - 2\varepsilon_{13}, \bar{k}_{11} = 2l_1 - \varepsilon_3^{-1} - \varepsilon_8, \bar{k}_{12} = 2l_2 - \varepsilon_5^{-1} - \varepsilon_7^{-1} - \varepsilon_9.$

**Proof.** According to the above discussion, choose the Lyapunov function candidate $V(t)$ as:

$$
\begin{aligned}
V(t) = {}& V_1(t) + V_2(t) + V_3(t) + V_4(t) + V_5(t) + V_6(t) + \frac{1}{2}\gamma_2^T(t)\gamma_2(t) + \frac{1}{2}\gamma_3^T(t)\gamma_3(t) \\
& + \frac{1}{2}\gamma_4^T(t)\gamma_4(t) + \frac{1}{2}\gamma_6^T(t)\gamma_6(t) + \frac{1}{2}\tilde{d}_1^{\,T}(t)\tilde{d}_1(t) + \frac{1}{2}\tilde{d}_2^{\,T}(t)\tilde{d}_2(t)
\end{aligned}
\tag{37}
$$

Combining (12), (17), (22), (27) and (30), we have:

$$
\begin{aligned}
\dot{V}(t) = {}& \dot{V}_1(t) + \dot{V}_2(t) + \dot{V}_3(t) + \dot{V}_4(t) + \dot{V}_5(t) + \dot{V}_6(t) + \gamma_2^T(t)\dot{\gamma}_2(t) + \gamma_3^T(t)\dot{\gamma}_3(t) \\
& + \gamma_4^T(t)\dot{\gamma}_4(t) + \gamma_6(t)\dot{\gamma}_6(t) + \tilde{d}_1^{\,T}(t)\dot{\tilde{d}}_1(t) + \tilde{d}_2^{\,T}(t)\dot{\tilde{d}}_2(t) \\
= {}& -k_1 e_1^T(t)e_1(t) + e_1^T(t)C_1(t)\gamma_2(t) - \frac{\varepsilon_1}{2}e_1^T(t)C_1(t)C_1(t)e_1(t) - k_2 e_2^T(t)e_2(t) \\
& + e_2^T(t)C_2(t)\gamma_3(t) + e_2^T(t)C_2(t)\tilde{d}_1(t) - \frac{\varepsilon_2}{2}e_2^T(t)C_2(t)C_2(t)e_2(t) \\
& - \frac{\varepsilon_3}{2}e_2^T(t)C_2(t)C_2(t)e_2(t) - k_3 e_3^T(t)e_3(t) + e_3^T(t)C_3(t)\gamma_4(t) \\
& - \frac{\varepsilon_4}{2}e_3^T(t)C_3(t)C_3(t)e_3(t) - k_4 e_4^T(t)e_4(t) - \frac{T_M(t)}{m}e_4^T(t)C_4(t)R(t)J^{-1}\tilde{d}_2(t) \\
& - \frac{\varepsilon_5}{2}\frac{T_M^2(t)}{m^2}e_4^T(t)C_4(t)R(t)J^{-1}J^{-1}R^T(t)C_4(t)e_4(t) - k_5 e_5^T(t)e_5(t) \\
& + \frac{e_5(t)\gamma_6(t)}{k_{b5}^2 - e_5^2(t)} - \varepsilon_6 \frac{e_5^T(t)e_5(t)}{\left[k_{b5}^2 - e_5^2(t)\right]^2} - k_6 e_6^T(t)e_6(t) + \frac{e_6(t)\alpha_1(t)J^{-1}\tilde{d}_2(t)}{k_{b6}^2 - e_6^2(t)} \\
& - \frac{\varepsilon_7}{2}\frac{e_6^T(t)e_6(t)}{\left[k_{b6}^2 - e_6^2(t)\right]^2}\alpha_1(t)J^{-1}J^{-1}\alpha_1^T(t) + \gamma_2^T(t)\dot{\gamma}_2(t) + \gamma_3^T(t)\dot{\gamma}_3(t) + \gamma_4^T(t)\dot{\gamma}_4(t) \\
& + \gamma_6^T(t)\dot{\gamma}_6(t) - l_1 \tilde{d}_1^T(t)\tilde{d}_1(t) + \tilde{d}_1^T(t)\dot{d}_1(t) - l_2 \tilde{d}_2^T(t)\tilde{d}_2(t) + \tilde{d}_2^T(t)\dot{d}_2(t)
\end{aligned}
$$

According to Lemma 1 and Young's inequality, there exist $\varepsilon_1 > 0, \varepsilon_2 > 0, \varepsilon_3 > 0, \varepsilon_4 > 0, \varepsilon_5 > 0, \varepsilon_6 > 0, \varepsilon_7 > 0, \varepsilon_8 > 0, \varepsilon_9 > 0$ such that:

$$
e_1^T(t)C_1(t)\gamma_2(t) \leqslant \frac{\varepsilon_1}{2}e_1^T(t)C_1(t)C_1^T(t)e_1(t) + \frac{\varepsilon_1^{-1}}{2}\gamma_2^T(t)\gamma_2(t)
$$

$$
e_2^T(t)C_2(t)\gamma_3(t) \leqslant \frac{\varepsilon_2}{2}e_2^T(t)C_2(t)C_2^T(t)e_2(t) + \frac{\varepsilon_2^{-1}}{2}\gamma_3^T(t)\gamma_3(t)
$$

$$
e_2^T(t)C_2(t)\tilde{d}_1(t) \leqslant \frac{\varepsilon_3}{2}e_2^T(t)C_2(t)C_2^T(t)e_2(t) + \frac{\varepsilon_3^{-1}}{2}\tilde{d}_1^T(t)\tilde{d}_1(t)
$$

$$
e_3^T(t)C_3(t)\gamma_4(t) \leqslant \frac{\varepsilon_4}{2}e_3^T(t)C_3(t)C_3^T(t)e_3(t) + \frac{\varepsilon_4^{-1}}{2}\gamma_4^T(t)\gamma_4(t)
$$

$$
- \frac{T_M(t)}{m}e_4^T(t)C_4(t)R(t)J^{-1}\tilde{d}_2(t) \leqslant
$$

$$
\frac{\varepsilon_5}{2}\frac{T_M^2(t)}{m^2}e_4^T(t)C_4(t)R(t)J^{-1}J^{-1}R^T(t)C_4(t)e_4(t) + \frac{\varepsilon_5^{-1}}{2}\tilde{d}_2^T(t)\tilde{d}_2(t)
$$

$$
\frac{e_5(t)\gamma_6(t)}{k_{b5}^2 - e_5^2(t)} \leqslant \varepsilon_6 \frac{e_5^T(t)e_5(t)}{\left[k_{b5}^2 - e_5^2(t)\right]^2} + \varepsilon_6^{-1}\gamma_6^T(t)\gamma_6(t)
$$

$$
\frac{e_6(t)\alpha_1(t)J^{-1}\tilde{d}_2(t)}{k_{b6}^2 - e_6^2(t)} \leqslant \frac{\varepsilon_7}{2}\frac{e_6^T(t)e_6(t)}{\left[k_{b6}^2 - e_6^2(t)\right]^2}\alpha_1(t)J^{-1}J^{-1}\alpha_1^T(t) + \frac{\varepsilon_7^{-1}}{2}\tilde{d}_2^T(t)\tilde{d}_2(t)
$$

$$
\tilde{d}_1^T(t)\dot{d}_1(t) \leqslant \frac{\varepsilon_8}{2}\tilde{d}_1^T(t)\tilde{d}_1(t) + \frac{\varepsilon_8^{-1}}{2}\dot{d}_1^T(t)\dot{d}_1(t)
$$

$$
\tilde{d}_2^T(t)\dot{d}_2(t) \leqslant \frac{\varepsilon_9}{2}\tilde{d}_2^T(t)\tilde{d}_2(t) + \frac{\varepsilon_9^{-1}}{2}\dot{d}_2^T(t)\dot{d}_2(t)
$$

According to (15), (20), (25) and (33), using the Lemma 1 and Young's inequality, there exist $\varepsilon_{10} > 0, \varepsilon_{11} > 0, \varepsilon_{12} > 0, \varepsilon_{13} > 0$ such that:

$$\gamma_2^T(t)\dot{\gamma}_2(t) \leqslant -\frac{\gamma_2^T(t)\gamma_2(t)}{\lambda_1} + \frac{\varepsilon_{10}}{2}\gamma_2^T(t)\gamma_2(t) + \frac{\varepsilon_{10}^{-1}}{2}\dot{\boldsymbol{\alpha}}_1^T(t)\dot{\boldsymbol{\alpha}}_1(t)$$

$$\gamma_3^T(t)\dot{\gamma}_3(t) \leqslant -\frac{\gamma_3^T(t)\gamma_3(t)}{\lambda_2} + \frac{\varepsilon_{11}}{2}\gamma_3^T(t)\gamma_3(t) + \frac{\varepsilon_{11}^{-1}}{2}\dot{\boldsymbol{\alpha}}_2^T(t)\dot{\boldsymbol{\alpha}}_2(t)$$

$$\gamma_4^T(t)\dot{\gamma}_4(t) \leqslant -\frac{\gamma_4^T(t)\gamma_4(t)}{\lambda_3} + \frac{\varepsilon_{12}}{2}\gamma_4^T(t)\gamma_4(t) + \frac{\varepsilon_{12}^{-1}}{2}\dot{\boldsymbol{\alpha}}_3^T(t)\dot{\boldsymbol{\alpha}}_3(t)$$

$$\gamma_6(t)\dot{\gamma}_6(t) \leqslant -\frac{\gamma_6^T(t)\gamma_6(t)}{\lambda_4} + \varepsilon_{13}\gamma_6^T(t)\gamma_6(t) + \varepsilon_{13}^{-1}\dot{\boldsymbol{\alpha}}_5^T(t)\dot{\boldsymbol{\alpha}}_5(t)$$

Moreover, from Lemma 2, the following inequalities hold:

$$\frac{1}{2}\sum_{i=1}^{3}\ln\frac{k_{b1i}^2}{k_{b1i}^2 - e_{1i}^2(t)} \leqslant \boldsymbol{e}_1^T(t)\boldsymbol{C}_1(t)\boldsymbol{e}_1(t)$$

$$\frac{1}{2}\sum_{i=1}^{3}\ln\frac{k_{b2i}^2}{k_{b2i}^2 - e_{2i}^2(t)} \leqslant \boldsymbol{e}_2^T(t)\boldsymbol{C}_2(t)\boldsymbol{e}_2(t)$$

$$\frac{1}{2}\sum_{i=1}^{3}\ln\frac{k_{b3i}^2}{k_{b3i}^2 - e_{3i}^2(t)} \leqslant \boldsymbol{e}_3^T(t)\boldsymbol{C}_3(t)\boldsymbol{e}_3(t)$$

$$\frac{1}{2}\sum_{i=1}^{3}\ln\frac{k_{b4i}^2}{k_{b4i}^2 - e_{4i}^2(t)} \leqslant \boldsymbol{e}_4^T(t)\boldsymbol{C}_4(t)\boldsymbol{e}_4(t)$$

$$\frac{1}{2}\ln\frac{k_{b5}^2}{k_{b5}^2 - e_5^2(t)} \leqslant \frac{e_5^T(t)e_5(t)}{k_{b5}^2 - e_5^2(t)}$$

$$\frac{1}{2}\ln\frac{k_{b6}^2}{k_{b6}^2 - e_6^2(t)} \leqslant \frac{e_6^T(t)e_6(t)}{k_{b6}^2 - e_6^2(t)}$$

Hence, we have:

$$\begin{aligned}\dot{V}(t) \leqslant &-k_1\boldsymbol{e}_1^T(t)\boldsymbol{e}_1(t) - k_2\boldsymbol{e}_2^T(t)\boldsymbol{e}_2(t) - k_3\boldsymbol{e}_3^T(t)\boldsymbol{e}_3(t) - k_4\boldsymbol{e}_4^T(t)\boldsymbol{e}_4(t) - k_5 e_5^T(t)e_5(t) \\ &- k_6 e_6^T(t)e_6(t) - \frac{1}{2}\bar{k}_7\gamma_2^T(t)\gamma_2(t) - \frac{1}{2}\bar{k}_8\gamma_3^T(t)\gamma_3(t) - \frac{1}{2}\bar{k}_9\gamma_4^T(t)\gamma_4(t) - \frac{1}{2}\bar{k}_{10}\gamma_6^T(t)\gamma_6(t) \\ &- \frac{1}{2}\bar{k}_{11}\tilde{\boldsymbol{d}}_1^T(t)\tilde{\boldsymbol{d}}_1(t) - \frac{1}{2}\bar{k}_{12}\tilde{\boldsymbol{d}}_2^T(t)\tilde{\boldsymbol{d}}_2(t) + D\end{aligned} \quad (38)$$

where $D = \frac{\varepsilon_{10}^{-1}}{2}\dot{\boldsymbol{\alpha}}_1^T(t)\dot{\boldsymbol{\alpha}}_1(t) + \frac{\varepsilon_{11}^{-1}}{2}\dot{\boldsymbol{\alpha}}_2^T(t)\dot{\boldsymbol{\alpha}}_2(t) + \frac{\varepsilon_{12}^{-1}}{2}\dot{\boldsymbol{\alpha}}_3^T(t)\dot{\boldsymbol{\alpha}}_3(t) + \varepsilon_{13}^{-1}\dot{\boldsymbol{\alpha}}_5^T(t)\dot{\boldsymbol{\alpha}}_5(t) + \frac{\varepsilon_8^{-1}}{2}\dot{\boldsymbol{d}}_1^T(t)\dot{\boldsymbol{d}}_1(t) + \frac{\varepsilon_9}{2}\tilde{\boldsymbol{d}}_2^T(t)\tilde{\boldsymbol{d}}_2(t) + \frac{\varepsilon_9^{-1}}{2}\dot{\boldsymbol{d}}_2^T(t)\dot{\boldsymbol{d}}_2(t)$, for $i = 1, 2, 3$

By defining $\lambda = \min\left\{k_1, k_2, k_3, k_4, k_5, k_6, \bar{k}_7, \bar{k}_8, \bar{k}_9, \bar{k}_{10}, \bar{k}_{11}, \bar{k}_{12}\right\}$, (38) can be written as:

$$\dot{V} \leqslant -2\lambda V + D \quad (39)$$

Thus the unmanned helicopter tracking error systems (6) can converge to the expected bounded range. □

## 5. Numerical Simulation

In order to verify the state-constrained flight-tracking controller designed in this paper. The parameters presented in Table 1 are selected as the physical parameters of unmanned helicopter and simulated by MATLAB/Simulink [28]:

**Table 1.** Parameters of unmanned helicopter.

| Parameter(unit) | Parameter Description | Parameter (Unit) | Parameter Description |
|---|---|---|---|
| $m = 8$ kg | quality of the helicopter | $J_{xx} = 0.26$ kg · m$^2$ | moment of rotation |
| $J_{yy} = 0.35$ kg · m$^2$ | moment of rotation | $J_{zz} = 0.29$ kg · m$^2$ | moment of rotation |
| $C_{ma} = 107$ (N · m/rad) | pitch moment intensity factor | $C_{mb} = 199$ (N · m/rad) | rolling moment intensity factor |
| $C_{MQ} = 0.0044$ (M · N$^{-\frac{1}{2}}$) | main rotor torque factor | $D_{MQ} = 0.6304$ (M · N$^{-\frac{1}{2}}$) | main rotor torque factor |
| $x_m = 0$ m | distance between the center of the main rotor and the x-axis of the helicopter's center of gravity | $z_m = 0.284$ m | distance between the center of the main rotor and the z-axis of the helicopter's center of gravity |
| $x_t = 0.915$ m | distance between the center of the tail and the x-axis of the helicopter's center of gravity | $z_t = 0.104$ m | distance between the center of the tail and the z-axis of the helicopter's center of gravity |

The initial states of the unmanned helicopter are:

$$\boldsymbol{P}(0) = [2\ 2\ 1.5]^T m, \psi(0) = -0.1 rad$$

The expected position and yaw of the unmanned helicopter are:

$$\boldsymbol{P}_d = [10cos(0.2t)\ 10sin(0.2t)\ 5 + 0.1t]^T m, \psi_d = 0.1sin(0.1t) rad$$

In order to verify the anti-disturbance ability of proposed control method, the disturbances of the unmanned helicopter are set as:

$$\boldsymbol{d}_1 = [1.5\sin(0.5t)\ 0.8\sin(0.4t)\ 1.3\sin(0.6t)]^T,$$
$$\boldsymbol{d}_2 = [0.5\sin(0.45t)\ 0.6\sin(0.5t)\ 0.6\sin(0.4t)]^T.$$

The simulation results of the unmanned helicopter are shown in Figures 2–10. The position tracking trajectories of the unmanned helicopter are given in Figure 2, which is a 3D figure, in which the helicopter could achieve the position-tracking task with good control performances. Since the unmanned helicopter is under the action of the full state-constrained flight-tracking controller, the velocities of the helicopter, accelerations generated by controllable join force and force variation rates generated by controllable join force are all constrained. Therefore, the positions of all three axes track the desired trajectories asymptotically with a relatively gentle trend, and maintain the tracking states in the subsequent process. The Figure 3 presents the position error curves on the three axes of the unmanned helicopter, which converge to the desired bounded range under the action of state constraint controller. Moreover, the maximum errors of position variables do not exceed $1 \times 10^{-2}$ m. Obviously, the helicopter has high control accuracy with the constructed anti-disturbance flight controller. The Figure 4 shows the unmanned helicopter attitude angular curves. All these helicopter intermediate control variables are also bounded and kept within a reasonable range which means, though the attitude angles are not the control target variables, these important variables are also in the bounded scopes. Hence, the results from this figure declare the helicopter could safely fly under our full state-constrained control scheme. The control inputs of the unmanned helicopter are shown in Figure 5. To suppress the influence of disturbances, the control inputs display the corresponding waves with the feasible scopes. Figures 6 and 7 represent the disturbance forces and disturbance moments $d_1(t), d_2(t)$ and their estimated curves.

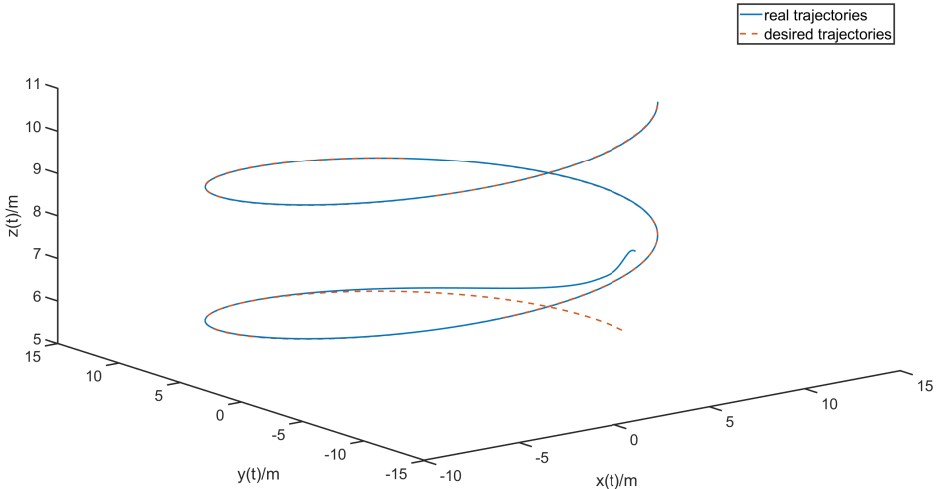

**Figure 2.** Three−dimensional image of position tracking trajectories of the unmanned helicopter.

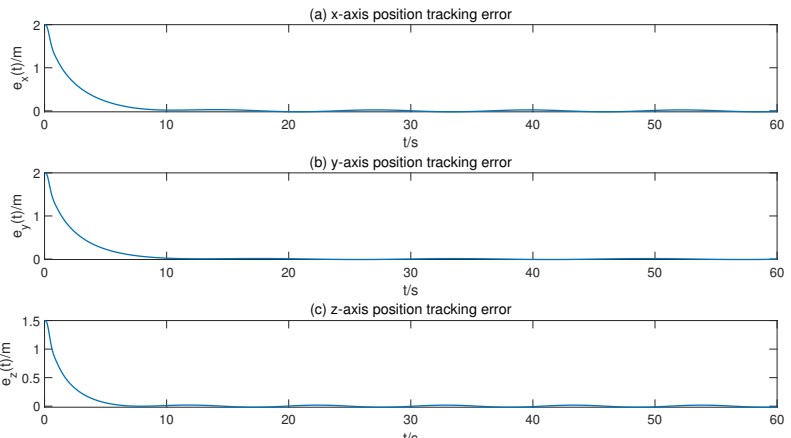

**Figure 3.** Position tracking error curves of the unmanned helicopter.

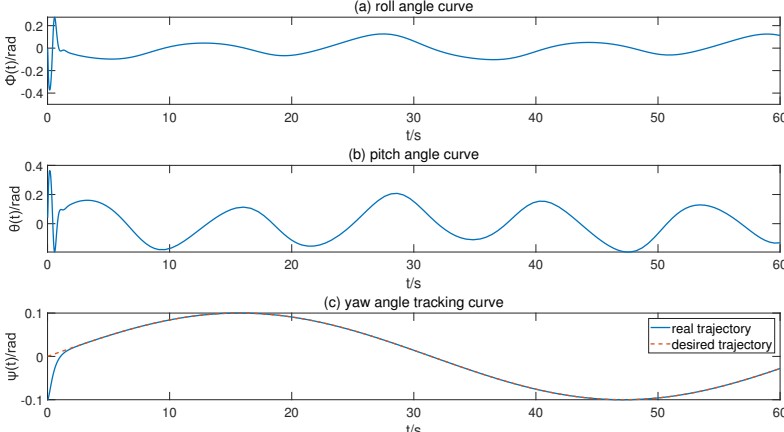

**Figure 4.** Attitude angle curves of the unmanned helicopter.

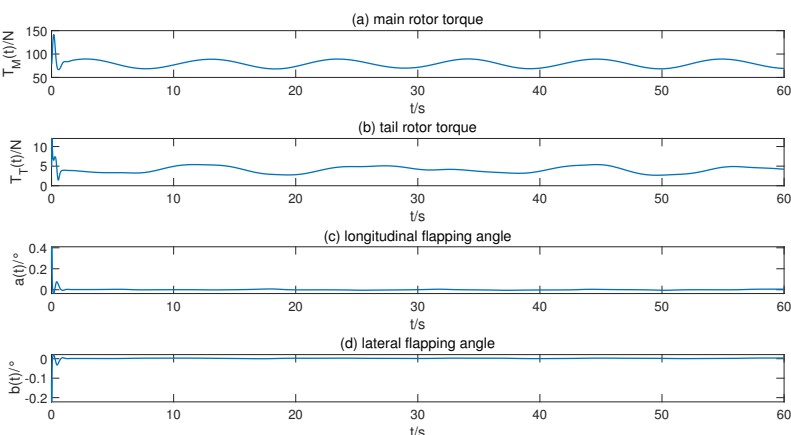

**Figure 5.** Control input curves of the unmanned helicopter.

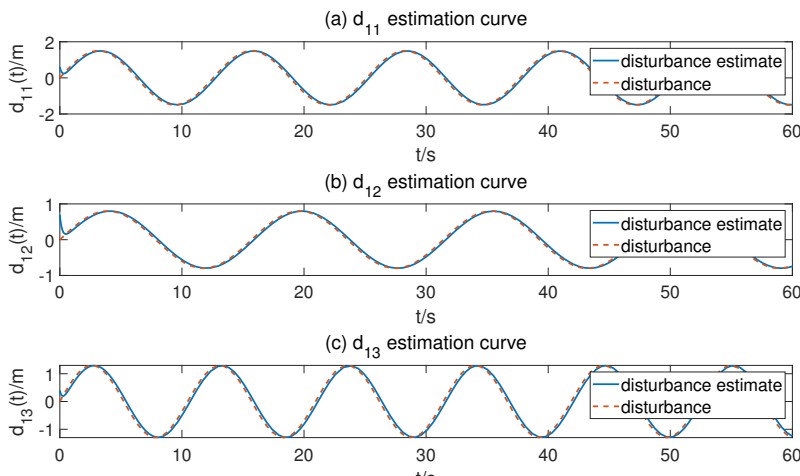

**Figure 6.** The curves of disturbance forces and their estimates.

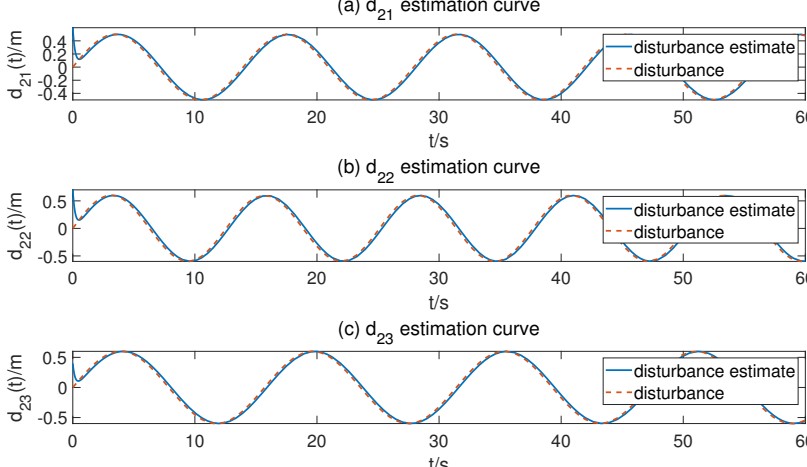

**Figure 7.** The curves of disturbance moments and their estimates.

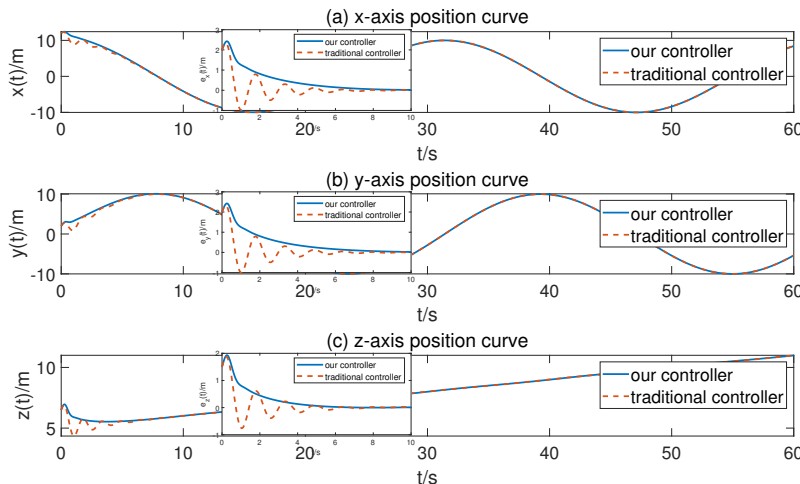

**Figure 8.** Position curves of the unmanned helicopter.

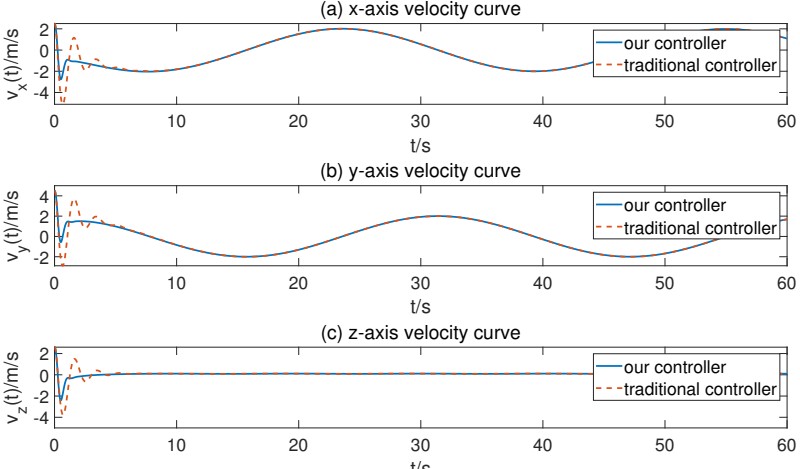

**Figure 9.** Velocity curves of the unmanned helicopter.

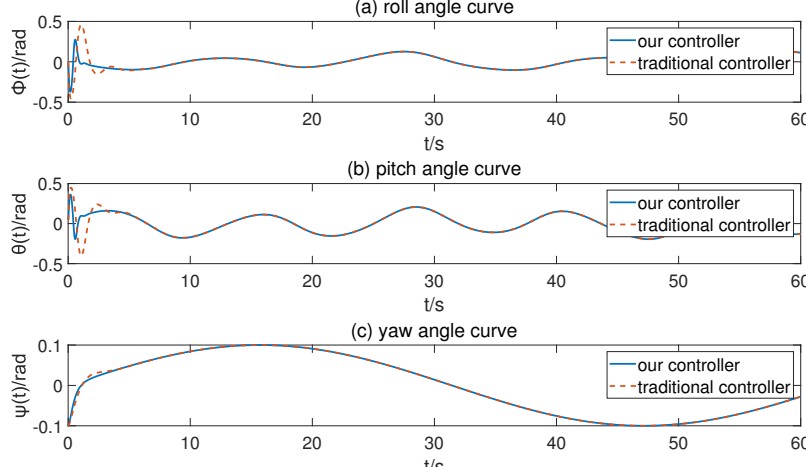

**Figure 10.** Attitude angle curves of the unmanned helicopter.

In order to demonstrate the advantages of our full state-constrained anti-disturbance control scheme, a typical nonlinear control method is applied for designing the flight-tracking controller without using the state-constrained control. The typical nonlinear control method is constructed by combining the nonlinear disturbance observer technology

and backstepping control method, which does not adopt the full state-constrained control scheme. The simulation results are shown in Figures 8–10, which represent the comparison of the unmanned helicopter position, velocity and attitude angle under the action of the full state-constrained flight-tracking controller and the backstepping controller, respectively, in which the helicopter systems have the same initial conditions. From the figures, it can be seen that the positions, velocities and attitude angles of the unmanned helicopter can be ensured to be constrained within the desired range under the action of the full state-constrained flight-tracking controller to avoid safety problems. Apparently, the curves of the positions, velocity and attitude angles have the smaller overshoot using our flight controller, which implies the helicopter has better control dynamic performance under the proposed flight control scheme.

## 6. Conclusions

In this paper, the anti-disturbance flight-tracking control problem is studied for unmanned helicopter under full state constraints and disturbances. Firstly, the input–output feedback linearization method is adopted for the unmanned helicopter system to reduce the complexity of controller design. Secondly, two nonlinear disturbance observers are used to estimate these system disturbances. Thirdly, a Barrier Lyapunov function method is combined with the backstepping control technology to construct the full state-constrained anti-disturbance flight-tracking controller. Then, the virtual control laws and their derivatives are estimated by the dynamic surface method to reduce the degree of "differential explosion" caused by nonlinear controller for the high-order system. Finally, the system stability is guaranteed by using the semi-global stability theory, and the effectiveness of designed controller is verified by numerical simulation.

**Author Contributions:** Conceptualization, Y.L. methodology, Y.L. and Y.H.; software, Y.H.; validation, Y.L. and D.L.; formal analysis, D.L.; writing—original draft preparation, Y.L. and Y.H.; writing—review and editing, D.L.; supervision, H.L.; funding acquisition, Y.L. and D.L. All authors have read and agreed to the published version of the manuscript.

**Funding:** This research was funded by the National Natural Science Foundation of China grant numbers 62103327 and 12101481, and the Project funded by China Postdoctoral Science Foundation grant numbers 2021MD703879 and 2022T150524.

**Data Availability Statement:** Not applicable.

**Conflicts of Interest:** The authors declare no conflict of interest.

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
