# Peer review of "Full State Constrained Flight Tracking Control for Helicopter Systems with Disturbances"

_aerospace, doi:10.3390/aerospace10050471_

Round 1

Reviewer 1 Report

This paper presents a state constrained anti-disturbance dynamic surface control method for the six-degree-of-freedom unmanned helicopter systems, wherein full-state constraints are addressed by employing BLF method in the framework of backstepping control. Overall, this paper is well-organized, which can be accepted. Meanwhile, some suggestions are shown as follows:

1.     The proposed control method is to enhance the dynamic performance of helicopters systems, the authors should give the details of how this works as a remark.

2.     The authors should explain the relationship between variables $x(t)$ and the other helicopter states, such as attitude angle, and so on.

3.     There are some grammatical problems in the current manuscript, for example, in Abstract line 7, “are used for constructed” should be “are used to constructed”.

4.     Spaces are usually required between English words, for example, in line 14 of page 1, vehicle (UAV).

5.     There are some typos in the current manuscript. Please double check the whole paper to improve the English.

6.     In the simulation section, the tracking error curves of comparison figures are suggested to be added, which would rich the advantages of proposed method.

Reviewer 2 Report

Refer to attachment for my comment and suggestion.

Reviewer 3 Report

The authors present a disturbance rejection methodology applied to a small-scale uninhabited helicopter. The paper provides lots of detail of the mathematical process associated with deriving the controller, but is missing some key ingredients for publication.

The introduction needs to evaluate previous research on disturbance rejection controllers, to define what is the current state-of-the-art in the field. In its current form, the paper just provides single-sentence summaries of previous works, which does not help to demonstrate the novelty of the current work. It is suggested that the introduction needs to be re-written to demonstrate the forefront of disturbance rejection techniques, and to explain the contribution that this paper makes to knowledge.

Although the proposed controller is outlined in detail, the 'benchmark nonlinear controller' is not explained. The authors needs to present this controller, and (with the help of a revised introduction) explain how it represents the current state-of-the-art in the field. 

The current results are minimal in volume and significance. The authors should consider what it is that they want to show from their simulation results, and identify the most effective way to present this. For example, are all states of equal importance to demonstrate controller performance? There is limited discussion of the current figures, which suggests that 12 figures are not needed to make the point(s) in the paper. Can the reader distinguish between controllers based on the visual difference between the figures, or is another measure more appropriate to use? Why are sinusoidal disturbances chosen? How do the controllers respond to more realistic disturbances (e.g. gust, which will be high frequency and not necessarily sinusoidal)? The results from this work therefore need to be revised.

Overall, the paper does not demonstrate sufficient novelty in its current form, and it lacks scientific rigour. Revisions should look to address these shortcomings. 

Round 2

Reviewer 3 Report

Thank you for taking the time to address my original comments. Unfortunately, the changes to the introduction do not sufficiently contextualise the work, and the results have not been updated to demonstrate the benefits of this controller over the state-of-the-art in the field, so my original review has not been addressed. Whilst I cannot recommend this for publication, my hope is that the editor will have enough information from other reviewers to make their decision.
